# Gender Differences among Healthcare Providers in the Promotion of Patient-, Person- and Family-Centered Care—And Its Implications for Providing Quality Healthcare

**DOI:** 10.3390/healthcare11040565

**Published:** 2023-02-14

**Authors:** Sarah Ashley Lim, Amir Khorrami, Richard J. Wassersug, Jame A. Agapoff

**Affiliations:** 1Faculty of Medicine, University of British Columbia, Vancouver, BC V6T 1Z4, Canada; 2Department of Psychiatry, John A Burns School of Medicine, University of Hawai’i at Manoa, Honolulu, HI 96813, USA

**Keywords:** patient-centered care, family-centered care, person-centered care, relationship-centered care, cognition, empathizing–systemizing theory, gender

## Abstract

The concept of “patient-centered care” (PCC) emphasizes patients’ autonomy and is commonly promoted as a good healthcare practice that all of medicine should strive for. Here, we assessed how six medical specialties—pediatrics, OBGYN, orthopedics, radiology, dermatology, and neurosurgery—have engaged with PCC and its derivative concepts of “person-centered care” (PeCC) and “family-centered care” (FCC) as a function of the number of female physicians in each field. To achieve this, we conducted a scoping review of three databases—PubMed, CINAHL, and PsycInfo—to assess the extent that PCC, PeCC, FCC, and RCC were referenced by different specialties in the medical literature. Reference to PCC and PeCC in the literature correlates significantly with the number of female physicians in each field (all *p* < 0.00001) except for neurosurgery (*p* > 0.5). Pediatrics shows the most extensive reference to PCC, followed by OBGYN, with a significant difference between all disciplines (*p* < 0.001). FCC remains exclusively embraced by pediatrics. Our results align with documented cognitive differences between men and women that recognize gender differences in empathizing (E) versus systemizing (S) with females demonstrating E > S, which supports PCC/PeCC/FCC approaches to healthcare.

## 1. Introduction

Over the last several decades, patient-centered care (PCC) has become an increasingly popular approach in Western medicine [1]. PCC emphasizes that healthcare providers (HCPs) consider each patient’s sociocultural background and autonomy when recommending and providing treatment [2,3,4,5,6]. This framework allows shared decision-making, with patient values and autonomy being central to treatment decisions [7,8,9]. Patients of diverse ages, socioeconomic status, gender, and pathology rate PCC as having above average importance to them in their interactions with physicians [10,11]. Thus, it has been praised as a standard that all HCPs should aspire to [5,12].

Accordingly, in the 1990s, the Picker Commonwealth Program was granted USD 9.7 million to bring PCC to the forefront of Massachusetts Hospital Health Services [13]. In 2001, the US Institute of Medicine (IOM, now the National Academy of Medicine) published *Crossing the Quality Chasm: A New Health System for the 21st Century* [3], which promoted how medicine should be practiced within a PCC framework. Moreover, a PubMed search on “patient-centered care” reveals a consistent rise in the number of citations related to PCC in the medical literature over the last 30 years (see Figure 1) with only a recent flattening of the curve [14]. According to Google Scholar as of October 2022, the IOM treatise has been cited over 7000 times.

Despite the substantial academic enthusiasm for the PCC approach, it lacks a uniformly agreed-upon definition [1]. Though there is some consensus on what PCC should or should not be, the literature is vague as to what are the most important elements in clinical settings. For example, the IOM promoted PCC as “providing care that is respectful of and responsive to individual patient preferences, needs, and values and ensuring that patient values guide all clinical decisions” [3]. Others describe it as enhancing the relationship between physician and patient, exploring the patients’ needs and wants and mutually agreeing on treatment decisions [1,4]. More recently, PCC has been broadly conceptualized as having the core values of empathy, respect, communication, and shared decision-making, with a focus on coordinated care among medical specialties [7,15,16].

To better understand the essence of PCC across medical specialties, one might explore what PCC is not. However, in trying to identify the opposite of PCC, we similarly find inconsistencies. Some authors have suggested that the opposite of PCC is volume-based care, which prioritizes revenue and efficiency for institutions and individual HCPs over patient satisfaction [17,18,19]. Others contrast PCC with paternalistic medicine, which limits patients’ role in treatment decisions with little regard to their preferences [20]. It is also thought that the opposite of PCC is disease-centered or evidence-based care [21]. This is a framework in which the precise pathology is considered key to selecting and undertaking treatment rather than the patients’ broader concerns [22,23].

PCC has additionally been distinguished from similar concepts, for example, person-centered care (PeCC). PeCC has been defined as prioritizing patient functionality in a medical decision over a patient’s emotional contentment and centers around long-term relationships rather than singular interactions in PCC [15,24]. Contrarily, others note that PCC and PeCC are generally considered equivalent in much of the world [1,25]. Interestingly, PCC is the more common term in North America and the UK, with PeCC being more popular in Europe [12,26]. It is also noted that PCC is more often used in the medical literature, whereas the nursing literature shows a greater preference for PeCC [12,25,26]. Nolte et al. (2020) note that the concept of PCC first emerged in the USA in the 1960s primarily within the context of general and family medicine. Within the nursing profession, proponents trace the concept’s origin back to Florence Nightingale [1].

In this paper, we explore several topics related to the concept of PCC. One goal is to assess the evolution and stability of engagement with PCC and derivative concepts among different medical specialties. These derivative concepts include person-centered (PeCC) care and two less commonly referenced care terms: relationship-centered care (RCC) and family-centered care (FCC) [1]. The concept of PCC within a selection of medical specialties has been recently addressed in the literature. Here, we extend the findings of those authors by investigating the volume of literature referencing these additional care concepts over the past two decades [14]. All of this is motivated by a desire to understand what characterizes and accounts for variation in how patient centricity is interpreted in various medical specialties. We explore what may account for differences in the references to these terms among medical specialties and the clinical implications that come from the ways in which PCC, PeCC, RCC, and FCC are engaged in the stated specialties.

The qualitative and quantitative evidence suggests that females have a greater capacity for empathy beyond cultural expectations of gender roles. For example, in affective empathy, females show higher emotional responsivity than men, and sex differences in empathy appear to be consistent and enduring throughout the lifespan [27]. This study aims to determine whether any correlations exist between the adoption of these concepts and changing gender ratios within specialties. As PCC focuses on the core values of empathy, respect, communication, and shared decision-making [7,15,16], we hypothesize that specialties with a greater percentage of female physicians would have greater engagement with PCC and its derivative concepts of care. If this is valid, then, as a specialty’s gender ratio changes, we envision shifts in engagement with the tenets of PCC.

Here, we examine the medical literature in six specialties—pediatrics, obstetrics and gynecology (OBGYN), dermatology, radiology, and neurosurgery—that reference PCC and similar concepts as a proxy for the integration of PCC in their clinical practice. This serves as an indirect way to assess each specialty’s engagement with the approach of PCC as well as with derivative approaches of care centricity.

These specialties were chosen due to their historical differences in gender ratios [13]. For decades, pediatrics and OBGYN have been consistently female-represented and have shown the greatest reference to PCC in their literature [14]. In contrast, orthopedic surgery, neurosurgery, and radiology continually show both a lower-than-average number of female physicians [28,29] and less reference to PCC in their literature [14]. Dermatology is an exception. Although the percent of female physicians in dermatology rose to 50.5% as of 2019, which was greater than the average of 42.7% across all specialties at that time [28], over time, the dermatology literature has made little nor increasing reference to PCC [14].

## 2. Materials and Methods

Data on gender composition of our selected medical specialties were taken from the Canadian Medical Association’s annual Canadian physician data from 2000 to 2019 [28]. We used these data instead of similar statistics collected by the American Medical Association for the USA because the American data span fewer years and are not collected annually. We found a high correlation between the Canadian and American data in number and trend of female physicians in our medical specialties of interest in the years where they overlap (see Figure 2). Therefore, for those years where American data are unavailable, we use the CMA data to represent the portion of females in the medical specialties in North America overall, extrapolating to the end of 2020.

Three databases—PubMed, CINAHL, and PsycInfo—were scoped to assess the extent that the literature in different specialties independently reference PCC, PeCC, FCC, and RCC. MeSH terms and keywords were used to search in PubMed, CINAHL, and PsycInfo. Papers published up to 1 January 2021 were included. PCC was also searched in Google Trends to assess any increase in popular, everyday use of the term.

Medical specialty-specific terms were searched using the “OR” function and then combined with the “AND” function to search for papers indexed under the term “patient-centered care” (see Appendix A for complete list of indexed terms). We included spellings of both centred/centered and searched the word string with and without a hyphen following the word “patient”. Only papers written in English were included.

All papers were imported into COVIDENCE (2022) and deduplicated via their automatic algorithm [30]. In COVIDENCE, papers were first separated into specialty. Vetting of papers then occurred through a review of literature in accordance with our exclusion and inclusion criteria (see Table 1). Papers went through two rounds of screening by two independent reviewers. First abstracts were reviewed. Papers were sorted into categories of “yes”, “no”, and “maybe” to be accepted for use as data. Papers labeled “yes” and “maybe” were subjected to full text review. Any disagreements between reviewers were negotiated by a third reviewer at both the abstract and full text review stage.

With these papers, we then examined the change in the number of publications referencing the term “PCC” within the literature for each specialty cross-referenced against the percent of female physicians in each specialty using a Pearson correlation test. These correlations were deemed significant with alpha set at 0.05.

The same process, as outlined above, was completed for the terms “PeCC”, “FCC”, and “RCC.” To gain a sense of the public’s acceptance and understanding of the PCC approach, we also searched PCC and PeCC on Google Trends from 2000 to the present.

## 3. Results

### 3.1. Evolution of Concepts of Care

As of June 2021, there were a total of 37,273 publications indexed in PubMed referencing the term “patient-centered care.” In the same period, there were 2785 references to “person-centered care,” 3434 references to “family-centered care”, and 269 references to “relationship-centered care”. PCC is the oldest of the terms, with the first reference in PubMed dated 1948. This was followed by FCC in 1953, PeCC in 1968, and RCC in 1995. Below, we explore how these terms have evolved in relation to each other since the IOM 2001 publication.

#### 3.1.1. Person-Centered Care

The concept of PeCC has seen progressive uptake in the medical literature over the last two decades, with the trajectory of growth matching in shape that of PCC (Figure 3). Up until the last four years, this growth has largely lagged behind PCC by four or more years. Recently, however, annual references to PCC have started to flatten out and even decline, whereas references to PeCC have continued to climb. If this trend continues, within another three years, PeCC will overtake PCC as the dominant care centricity term in the academic literature.

As seen in Figure 4, the fields of pediatrics and OBGYN have referenced PeCC in their publications far more than the other medical specialties we examined. In contrast, orthopedics and radiology have both had fewer references and slower growth in published references to PeCC (Figure 4). There were so few references to PeCC in the dermatology and neurosurgery literature that we have not included them on the graph.

#### 3.1.2. Relationship-Centered Care

The concept of RCC had a major academic debut in 2006 in a set of papers in the *Journal of General Internal Medicine* [31,32,33]. Those three papers have, respectively, received 679, 269, and 208 citations. RCC, nevertheless, remains a rarely referenced care concept outside of internal medicine. It has shown some increase in citations over the last eight years but in a linear rather than exponential fashion. It has not seen an increase in reference in any of the specialties we examined.

#### 3.1.3. Family-Centered Care

As shown in Figure 5, the term “FCC” has been exclusively adopted by pediatrics compared to the other medical specialties. Reference to FCC shows exponential growth in the pediatric literature since the early 2000s. In contrast, there has been no growth for this concept in the other specialties we examined.

#### 3.1.4. Patient-Centered Care

Even though “PCC” is the oldest and most frequently referenced term out of the four we examined, the annual citations of the term increased in a linear rather than exponential fashion from approximately 2007 to 2017. Since then, citations have plateaued at ~44 per year (Figure 3). In the same period, PeCC citations have increased at a rate of 69 papers per year, FCC at 27 papers per year, and RCC at only 3 papers per year. This is evidence of both the general acceptance of the term “patient centered care” in the medical literature and a rising preference for the more encompassing label “PeCC.” The shift in terminology from PCC to PeCC, however, does not show evidence of increased engagement with patient/person-centric approaches of medical specialties outside of the ones that have driven the original rise of citations relating to PCC in the medical literature.

Collectively, these data show substantial variations in the growth and evolution of reference to centricity-in-care language among specialties. Pediatrics and OBGYN have taken up the derivative concepts of PeCC and FCC, with signs of these terms replacing PCC. However, in the other disciplines we examined, these derivative concepts of care have shown little or no uptake, even though their names imply a broader and more inclusive reach than PCC [1].

Many of the medical papers we examined used the term “patient centered care” only once with no discussion or citations to indicate how the approach is precisely understood or operationally applied. This suggests that the term has been widely accepted in the medical literature despite little evidence of engagement with principles of patient centricity in clinical practice.

### 3.2. How Popular Is PCC and PeCC with the Public

As evidence for the general popularity of the term “patient centered care” beyond the medical literature, a Google search on “patient centered care” yielded ~1.69 billion hits as of 15 January 2022. The same search on “PeCC” approximately equals the number (at ~1.7 billion).

A Google Trend analysis shows PCC as the more common term in 2016–2017. In 2021, there was an average of 38 Google searches worldwide for PCC per week and 17 on PeCC. However, in 2016, they averaged 31 and 6 searches, respectively. This shows a higher slope for PeCC, indicating its use is growing more rapidly.

When searching in just the United States, Google Trends reveals a consistently lower level of PeCC than PCC since 2004—the time at which Google Trends first became publicly accessible. In fact, from January 2004 to March 2007, there were no searches in the USA for PeCC.

### 3.3. Gender, Specialties, and Engagement with PCC

In testing our primary hypothesis that engagement with PCC is gender-correlated, we examined the relationship between the percentage of female physicians in our sample of six medical specialties and the extent to which the literature for those specialties reference PCC over the past 20 years.

In this time frame, the number of publications referencing PCC was strongly correlated with the percentage of female physicians in pediatrics, OBGYN, dermatology, radiology, and orthopedics with correlation coefficients of 0.86, 0.91, 0.87, 0.90, and 0.92, respectively (see Table 2). All these correlation coefficients were statistically significant with *p* < 0.00001. In neurosurgery, the correlation coefficient between the number of PCC publications and the percentage of female physicians was 0.44, with *p* = 0.051.

Coincidentally, this gendered difference was present in the IOM’s Committee on Quality of Health Care in America, which produced the seminal 2001 treatise promoting PCC. Notably, all members of their study staff that produced the document were female. We have seen a similar pattern in other organizations that mention PCC and PeCC in their mission statements. For example, four out of five of the current board directors on the Institute of Patient- and Family-Centered Care https://www.ipfcc.org/ (accessed on 1 October 2022) are female, as are the rest of its 12-member team.

## 4. Discussion

PCC is a popular approach in both the medical literature and the public [1]. However, a positive perspective on PCC is not necessarily a reflection of how well the concept is understood by the public nor how extensively its principles are executed in clinical practice. Our data suggest that the term “PCC”, and to a lesser extent “PeCC”, is gaining support and is strongly adopted by specialties with a higher representation of female HCPs.

The adoption of PCC and PeCC, without considering alternative approaches of care (such as RCC), may result in lost opportunities to improve healthcare. It has already been demonstrated that different stakeholders in healthcare—i.e., clinicians, hospital management, and the lay public—understand these terms differently [26]. Our historical analysis of the medical literature suggests that major medical specialties have not engaged with these approaches to the same extent. It should be acknowledged that although a specialty may rarely mention PCC or PeCC in its academic literature, this does not imply that they are opposed to the concepts.

Even now, despite widespread support, PCC lacks a unified definition. Scholl et al. (2014) [32] found 417 articles offered general definitions and endorsements for patient-centered healthcare yet did not have a pragmatic and consistent consensus about what embodies PCC. As concluded by Nolte and Kitson [1,26] in their 2020 review, although “different stakeholders all agree that patient- or person-centeredness is important, the concept very much remains subject to debate, with different perspectives attaching different meanings and with different implications.” Outside of RCC, our data confirm the hypothesis that these differences tightly correlate with variances in the percentage of females in each specialty. For the literature in specialties where females are more represented, such as OBGYN and pediatrics, PCC was a topic of discussion well before the IOM’s 2001 paper. Indeed, reference to PCC climbed fast in the overall medical literature in the 20 years following that publication; however, the rank order of the specialties’ engagement with the concept did not change [14]. 

Our analysis of references to centricity-of-care terms derived from PCC, namely, PeCC and FCC, closely match what we saw with PCC. Once again, the two primarily female-represented medical specialties overwhelmingly account for the rise in citations to these approaches in the medical literature. The disciplines that first embraced PCC are also those that discuss and champion the derivative healthcare centricities of PeCC and FCC. In contrast, the uptake of the concept of RCC is so small that we do not have enough data to examine gender or disciplinary difference in the literature we reviewed referencing that term.

One implication of the similar trajectories for both “PeCC” and “FCC” in relation to the parent term “PCC” is that the newer terms have not led to a significant expansion of uptake in the medical specialties that are less female-represented. The terms PeCC and FCC were promoted, in part, to further improve healthcare and increase the reach of the original concept of PCC [12]. Our data suggest that changing the language has not made it more impactful, given the limited uptake of the derivative terms across medical specialties.

An underlying problem seems to be that the terms “FCC”, “RCC”, and “PeCC” are no more exact, precise, nor descriptive than “PCC” itself. Indeed, Hughes, Bamford, and May (2008) identified different types of centricities in clinical care, including PCC, FCC, PeCC, and RCC and concluded that these terms lack much difference [34]. In other words, the key themes of each term could be used to describe one another. In reviewing these terms, Starfield (2011), who favored PeCC over PCC, defined PeCC as “focusing on the entire person” [24]. A report by the World Health Organization (WHO) (2015) additionally stated PeCC “consciously adopts individuals’…perspectives around the comprehensive needs of people rather than individual diseases” [35]. However, a comprehensive approach to the patient is a core feature of PCC as well [4]. Proponents of distinguishing PeCC from PCC argue that it differs from PCC [1,24,36]. One explanation is that it focuses on the meaning and emotional fulfilment a patient may get from their treatment decision and empathetic relationship with their physician [37]. Supposedly PCC focuses more on functionality [1] and does not necessarily prioritize the person’s emotion-driven goals and needs [36,38]. This may be philosophically valid and an insightful distinction with positive implications for clinical care. However, our data suggest that it is not reflected in any change in the engagement with the concepts in male-represented specialties.

A shift in the language from PCC to PeCC may be seen by many as enriching the scope and advancing the agenda of those honorably working to improve the healthcare system. As noted by Nolte, Merker, and Anell (2020), preference for the term “PeCC” shows up particularly in nursing, which has historically been highly represented by females compared to medicine [1]. However, without a unified definition, and a wider adoption by stakeholders, “PCC” will likely remain the most widely adopted approach, which is in keeping with its dominance in the medical literature.

There are other ways of documenting the disciplinary divide in reference to PCC vs. PeCC. For example, if one looks at references to the word strings “patient-centred *medical* care” and “patient-centred *nursing* care” in Google Scholar, the former is 53% more common (N = 620 for nursing versus N = 951 for medicine). If one just looks at the last 10 years, the references for “patient” paired with “medical” climb to 80% more than references to “nursing”. This reiterates the point shown in Figure 2 that references to PCC have been growing in the academic healthcare literature overall, which may be driven proportionately more by medical rather than nursing literature.

If one then performs the same analysis as above for “person-centred *medical* care” versus “person-centred *nursing* care”, the word string with “nursing” embedded in it shows up 4X more often than with the word “medical” in the string (N = 247 and N = 63 for nursing versus medicine, respectively). Furthermore, if we again restrict the analysis to the last decade, the ratio shifts by a mere 1.5%, indicating only a slight rise in pairing PeCC with “nursing” over pairing it with “medical”. Overall, this shows that medicine, in contrast to nursing, has had far less engagement with “person-centred” language, and that has not changed in the last ten years.

Two other gender correlates may contribute to the general rise in references to both PCC and PeCC in the healthcare literature. As more nurses are publishing papers, we would expect more references to PeCC, as shown in Figure 2. In addition, for the last half-century in North America, there has been a continued rise in the number of graduated female physicians, with now half of all medical students being female [29]. They too have contributed to the rise in academic papers promoting patient- and person-centric healthcare.

Finally, the pediatrician Barbara Starfield, in her seminal 2011 paper “Is patient-centered care the same as person-focused care?” [24], makes a case for PeCC over PCC in primary care. She stresses the need for a long-term patient–provider relationship that goes beyond focusing on the specifics of the disease and takes into consideration the person’s experience of living with a disease. Starfield notes this is particularly relevant in the context of chronic illness, complex care needs, and/or debilitating conditions. Such patients have, by definition, healthcare challenges that cannot be easily cured. Yet, they can benefit from psychological support without being labeled a “patient” [24].

### Gender, Cognition, and Healthcare

Our data suggest that, from its origins to the current day, the approach of PCC and its derivative concept of PeCC have been promoted and engaged with by women more than by men.

This raises the question of why there is such a gender divide in patient-centric language in healthcare. The following comments are speculative but explore underlying factors that may account for gender differences in the patient-centric medical literature and practice. We acknowledge that our language here is simplistic and binary. It does not consider diversity in gender expression beyond a male/female dichotomy. However, the medical literature that is at the basis for this discussion is historically partitioned that way, so—without endorsing the binary—our discussion is framed in that language.

The divide between females and males in engagement with PCC/PeCC may be explained by sex-specific differences in the nonverbal and verbal communication skills between men and women. This may contribute to why patient centricity appears to be more relevant in female dominated specialties.

One of the most established differences between men and women relates to visual cognition and the ability to read facial expressions. As shown by Baron–Cohen and others [39,40,41,42], women are better able to accurately read facial expressions and the emotions behind them. Women have consistently proven to be both more aware of facial expressions and their meaning in various well-designed and well-controlled studies [43].

Understanding what another individual is saying is more fully comprehended if the speaker is facing the other person. What is commonly referred to as “eye contact” is an opportunity for the viewer to assess the valence in the facial expressions of the person before them. It thus follows that eye-contact is often presented as a key element in patient-centered care [44,45]. It also provides a chance to assess pupil dilation as a measure of interest and trust [46,47,48].

This “reading of the mind in the eyes” [41] is a prelude to and another stereotypical sex difference in the empathizing (E) vs. systemizing (S) classification. Women demonstrate E > S on various psychometric tests, which first and foremost require the accurate assessment of the emotional state of the person they are empathizing with [40,42].

## 5. Clinical Implications

In a practical sense, it is difficult to expect all HCPs to master both the E and S ends of the cognitive spectrum, but it is reasonable to expect them to recognize their personal limitations and try to improve them. PCC/PeCC and its derivatives take time to master [1,24], but so does mastering modern surgery with, say, the da Vinci robotic surgical system. Much of the extended life expectancy that patients now have can be credited to advances in the instrumentation and techniques provided by individuals who have exceptionally good hands-on, visual–spatial cognitive skills. Therefore, good medicine requires HCPs skilled with both S and E cognitive styles.

As gender proportions shift and more women enter medicine, we anticipate more changes to the medical culture that may arise. Adapting to these changes to deliver the most patient/person-centered and analytically sound medical care should be the aspiration of all healthcare providers. Some medical specialties, by their very nature, may lean on individuals innately proficient in S or E approaches. So, despite any gender differences that may have been found in this research, the path to better medicine and healthcare is one that concurrently maximizes both S and E skillsets to meet the patients’ clinical needs.

## 6. Limitations

Although we have framed our discussion around gender differences in the literature, other factors may impact the appropriation and implementation of patient-centered models of care in specific specialties, such as the age and cultural background of physicians. Many physicians will not fit into the sex-specific stereotypes we have described. In addition, we used the medical literature to infer difference in PCC/PeCC clinical practices among a small sample of medical specialties. Using literature references to the terms PCC, PeCC, RCC, and FCC as a proxy to clinical practice styles is clearly the biggest limitation to this study. We also acknowledge that the definitions of care types may differ across papers or specialties, which may impact intergroup differences and trend data. It should also be noted that publication trends across specialties may account for some differences in the relative number of papers on PeCC, RCC, and FCC. Another limitation is the novel reporting of Google Trend data, the reliability of which is not independently verifiable. Moreover, using data for just six medical specialties to make inferences about gender difference across all of medicine yields, at best, hypotheses that need larger data sets to be considered firm conclusions. Furthermore, we analyzed only references published in the English language. Lastly, our data for the percentage of female physicians in each medical specialty were Canadian extrapolated beyond USA data as the former covered a shorter period of time.

## 7. Conclusions

Patient-centered care is the principle that promotes shared decision-making between healthcare providers and patients, with patients’ sociocultural background and autonomy being central to the treatment plan. The implementation of PCC significantly improves patient-reported outcomes. Similar concepts such as PeCC, FCC, and RCC have been introduced in the past few years but have been adopted inconsistently among various medical specialties.

As predicted, our data suggest that differences in engagement with PCC (and its derivative constructs of PeCC and FCC) strongly correlate with the percentage of female HCPs in each specialty. Although reference to these concepts in the medical literature has increased in the last 20 years, the relative uptake of the terms in different fields has not changed; the concepts continue to be predominantly associated with female-represented medical specialties.

We found sex-specific cognitive differences between men and women are reflected in gender ratio differences in medical specialties. Women are faster and more accurate at recognizing facial expressions and other non-verbal emotional expressions than men. The ability to recognize emotions is important in affective empathy [27]. Such a gender difference is consistent with the variance in reference to PCC/PeCC/FCC in the literature for different medical specialties.

In the clinical setting, E and S behaviors reflect strengths and skills that have different values in different medical specialties. As such, the most comprehensive care would incorporate both cognitive styles regardless of a physician’s gender or whether they actively promote PCC/PeCC approach in their academic sphere. Shifts in medical student gender ratios, along with academic endorsements of PCC-related concepts, will likely continue to influence the evolution of healthcare.

## Figures and Tables

**Figure 1 healthcare-11-00565-f001:**
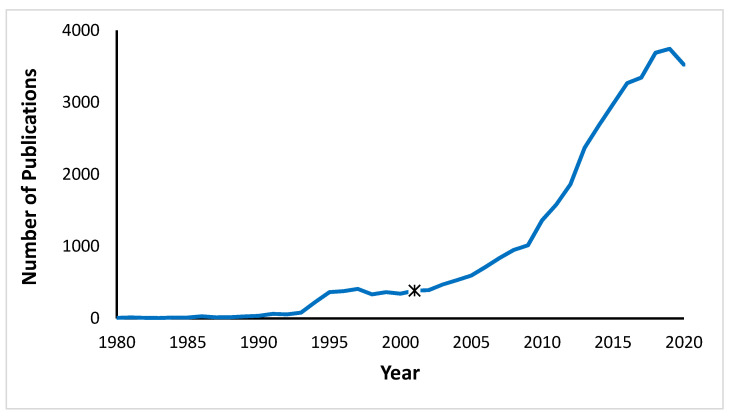
The number of papers over time indexed under “patient-centered care” in PubMed. Figure taken from Lim et al. [14]. Asterix on graph represents the year when the IOM published *Crossing the Quality Chasm* [3].

**Figure 2 healthcare-11-00565-f002:**
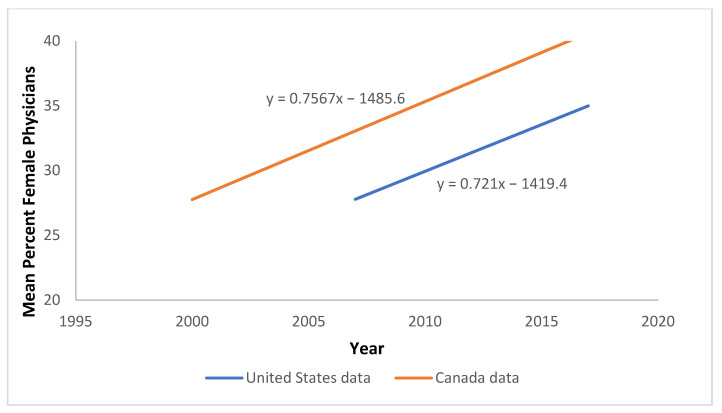
The regression model presenting the percentages of Canadian and American female physicians across the six medical specialties over time.

**Figure 3 healthcare-11-00565-f003:**
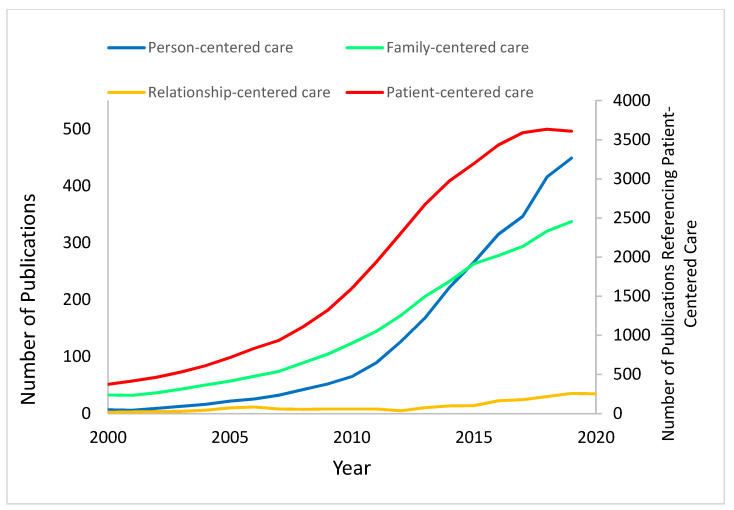
Three-point moving average regression analysis of the use of the term “family-centered care”.

**Figure 4 healthcare-11-00565-f004:**
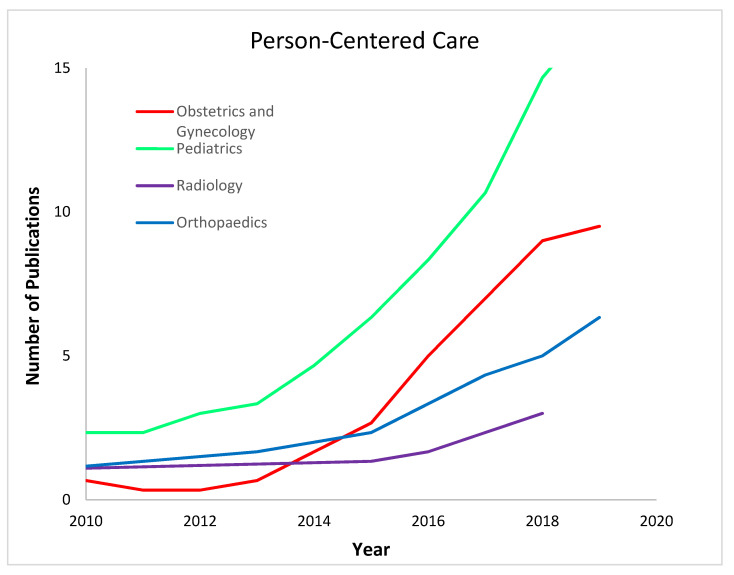
Three-point moving average regression analysis of the use of the term “person-centered care”.

**Figure 5 healthcare-11-00565-f005:**
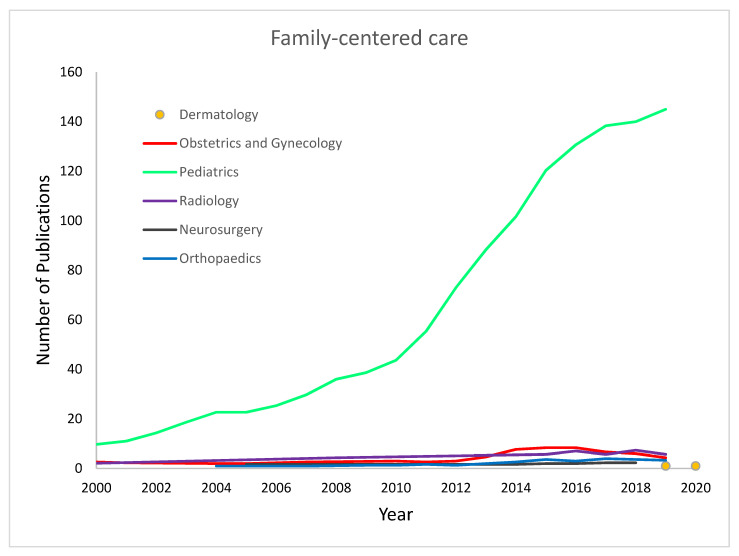
Three-point moving average regression analysis of the use of the term “family-centered care”.

**Table 1 healthcare-11-00565-t001:** Inclusion and exclusion criteria used to screen literature.

Inclusion Criteria	Exclusion Criteria
The paper must be in the field of medicine.	For neurosurgery and orthopedics, the issue being discussed in the paper must be the surgery procedure or consultation, therefore exclude any papers that only talk about perioperative- or postoperative-related interventions. This is because perioperative and postoperative interventions are often performed by nursing staff rather than the surgeons.
The healthcare practitioners in the paper must be physicians or surgeons.	Any paper that highlights PCC (or its derivatives) in the context of more than one of the six disciplines and does not discriminate between the use of PCC in these different disciplines.
PCC (or its derivatives) must be a main focus of the paper (mentioned at least once in the abstract and more than once in the full text).	If the paper identifies a single disease/class of diseases and discusses treatment modalities and efficacy of treatments, then it should be excluded.
The paper must include talk about clinical application of PCC (or its derivatives), rather than discuss only the language.	If no abstract is present to be able to screen, and no full text can be found online.
The primary specialty being discussed in the literature must be in one of the six specialties.	If the paper includes a healthcare multidisciplinary team.

**Table 2 healthcare-11-00565-t002:** The correlation between reference to “patient-centered care” in each medical specialty and the percentage of female physicians in those specialties from 2000 to 2019.

Medical Specialty	Percentage of Female Physicians in Years 2000–2019 (Mean)	Number of Papers Referenced “Patient-Centered Care” in Years 2000–2019 (Mean)	Correlation Coefficient (0–1)
Pediatrics Obstetrics and Gynecology Dermatology	41.3–62.2	2–43 (31)	0.86
	30.6–61.8	2–50 (19)	0.91
	35.1–50.5	0–8 (2)	0.87
Radiology Orthopedics	21.6–31.6	1–23 (8)	0.90
	5.4–12.6	0–30 (8)	0.92
Neurosurgery Medical specialty Pediatrics Obstetrics and Gynecology	5.6–10.6	0–8 (2)	0.44
	Percentage of female physicians in years 2000–2019 (mean)	Number of papers referenced “patient-centered care” in years 2000–2019 (mean)	Correlation coefficient (0–1)
	41.3–62.2	2–43 (31)	0.86
	30.6–61.8	2–50 (19)	0.91
Dermatology Radiology	35.1–50.5	0–8 (2)	0.87
	21.6–31.6	1–23 (8)	0.90

## Data Availability

No new data were created or analyzed in this study. Data sharing is not applicable to this article.

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
