# Peer review of "Gender Differences among Healthcare Providers in the Promotion of Patient-, Person- and Family-Centered Care—And Its Implications for Providing Quality Healthcare"

_healthcare, 2023, doi:10.3390/healthcare11040565_

Round 1

Reviewer 1 Report

The study fits the scope of Healthcare, but the link and justification between the different themes under study should be properly made. The study has some strengths, but unfortunately, they do not outweigh its limitations (including some methodological options that contributed to the limitations of the study). In the methods section data/statistical analysis should be explained. In the results section, results per database should be added. It is unclear whether a study is duplicated,  by mentioning two or more expressions under analysis.

There is a similar article published: Lim SA, Khorrami A, Wassersug RJ. Twenty years on - has patient-centered care been equally well integrated among medical specialties? Postgrad Med. 2022 Jan;134(1):20-25. doi: 10.1080/00325481.2021.2009237. 

Authors should avoid plagiarism and ensure that the content between the studies are different.

Author Response

We are unclear what "methodological options" the reviewer referred to as limiting. We noted that the other reviewers found the methods and results sections as clearly articulated and sound.

We benefited from the comment on plagiarism and reviewed are full manuscript. The reviewer is correct that Figure 1 needed to cite our earlier paper Lim et al. Other than this oversight, we found no evidence that we plagarized are own work or others. Our previous work focused on discipline differences, whereas the current work focuses on gender differences. The methodological approach is similar but the question is different. Of course, we appreciate your careful review of similar work.

Reviewer 2 Report

This is an interesting and well-designed study.

It demonstrates that literature on patient-centred care is different among different medical specialties. Moreover, it shows that the prevalence of papers on this topic is closely related to the presence of lady doctors in each spedialty. Finally it gives an intersting explanation of this observation, based on cognitive differences between men and women.

I have no major concerns about its publication, but some ameliorations and corrections could be helpful.

Major observations

1. Given the importance of available articles on patient-centred care in this study, I think that the percentage of these articles in each specialty should be given, if possible. Indeed the possibility exists that a small number of publications in a given specialty is due to an overall smaller number of articles in that specialty.

2. Suppl Table 1 gives core information. It should be inserted in the main text and not in supplementary meterials. 

3. The authors should mention the possibility that confounding factors exist, e.g. age and ethnicty of doctors in different specialties 

Minor observations

1. Line 5: probably "and" should be placed between "Wassersug" and "James"

2. Lines 124-126: this phrase is somehow unclear

3. Table 1, third item "PCC must be....": only PCC or also its derivatives?

4. Table 1, fifth item "The primary physician...": this is unclear

5 line 148: please specify the statistcal package you used

6. Line 213: probably, singular ("the approach") sounds better

7. Lines 378-397: this part can be omitted since the discussion is quite long

8. Line 399 "it difficult"; I guess the authors mean "it is difficult"

9. Lines 402-405: this phrase is somehow unclear

10. Line 428: could UK data be of some interest? A mention should be made

11. Line 433: "thus been praised": it is not clear

Author Response

Major observations

  1. Given the importance of available articles on patient-centred care in this study, I think that the percentage of these articles in each specialty should be given, if possible. Indeed the possibility exists that a small number of publications in a given specialty is due to an overall smaller number of articles in that specialty. 

Given the number of different journals, indexing platforms, and databases available, it would be difficult to estimate the total number of publications for a given specialty and determine whether the ratios of articles were similar when compared across specialties. To address this potential limitation we have added the following sentence to the limitations section:

"It should also be noted that publication trends across specialties may account for some differences in the relative number of papers on PeCC, RCC, and FCC." 

  1. Suppl Table 1 gives core information. It should be inserted in the main text and not in supplementary meterials. 

This table has been added to the main text and relabeled as "Table 2"

  1. The authors should mention the possibility that confounding factors exist, e.g. age and ethnicty of doctors in different specialties 

The authors have addressed these possible confounding factors with the following sentence added to the limitations section.

"Although we have framed our discussion around gender differences in the literature, other factors may impact the appropriation and implementation of patient-centered models of care in specific specialties such as the age and cultural background of physicians."

Minor observations

  1. Line 5: probably "and" should be placed between "Wassersug" and "Jame"

Corrected

  1. Lines 124-126: this phrase is somehow unclear. This is now line 121-123 in the revision.

The sentence has been updated to read as follows:

"Therefore, for those years where American data is unavailable, we use the CMA data to represent the portion of females in the medical specialties in North America overall, extrapolating to the end of 2020."

  1. Table 1, third item "PCC must be....": only PCC or also its derivatives?

Updated to: "PCC (or its derivatives)"

  1. Table 1, fifth item "The primary physician...": this is unclear

Changed to: "primary specialty"

5 line 148: please specify the statistcal package you used

Our paper has no complex statistics; e.g., no modelling, multi-variate statistics, or T-tests. The few stats we have are in basic Excel software, which we feel is not necessary to report.

  1. Line 213: probably, singular ("the approach") sounds better

Updated

  1. Lines 378-397: this part can be omitted since the discussion is quite long

Deleted

  1. Line 399 "it difficult"; I guess the authors mean "it is difficult"

Updated

  1. Lines 402-405: this phrase is somehow unclear

Deleted

  1. Line 428: could UK data be of some interest? A mention should be made

All English language studies, including those from the UK, were included.

  1. Line 433: "thus been praised": it is not clear

Updated

Reviewer 3 Report

Dear authors,

Thank you for the opportunity to review the manuscript. I have some suggestions as follows:

- Title: I would suggest adding wording like among healthcare providers/ subspecialties/professionals because reading this title suggested thinking that the manuscript examined the angle of patients' gender inequity.

- Abstract: Please stated the clear aim of the review, and what type of study design would this manuscript belong to a systematic review, narrative review, rapid review, etc.

- Another issue that might need to be considered, as mentioned in the discussion part, is the definitions of each care type. Some articles used different definitions for the same care types, particularly, patient-centred care and person-centred care. Thus, it should be aware when comparing the results and reporting the trend of each care type and could be a flaw in the manuscript.

- Introduction:

       - please stated the clear aim and add the references for the statement 'specialities with a greater percentage of female physicians
would have greater engagement with PCC and its derivative concepts of care.' (Line 93).

      - Did each speciality demonstrate the differences in PCC, PeCC, RCC and FCC? could it possible that the factor affecting the care be specialities instead of genders?

      - would it be reliable to report google trends in a scientific paper?

Best regards,

Reviewer

Author Response

-Title: I would suggest adding wording like among healthcare providers/ subspecialties/professionals because reading this title suggested thinking that the manuscript examined the angle of patients' gender inequity.

The title has been updated to... 

Gender Differences Among Healthcare Providers in the Promotion of Patient-, Person- and Family-Centered Care…And its Implications to Providing Quality Healthcare

-Abstract: Please stated the clear aim of the review, and what type of study design would this manuscript belong to a systematic review, narrative review, rapid review, etc.

The following sentence was added to the abstract...

To achieve this, we conducted a scoping review of three databases—PubMed, CINAHL, and PsycInfo—to assess the extent that PCC, PeCC, FCC, and RCC were referenced by different specialties in the medical literature.  

-Another issue that might need to be considered, as mentioned in the discussion part, is the definitions of each care type. Some articles used different definitions for the same care types, particularly, patient-centred care and person-centred care. Thus, it should be aware when comparing the results and reporting the trend of each care type and could be a flaw in the manuscript.

The authors added the following statement to the limitations section:

We also acknowledge that the definitions of care types may differ across papers or specialties, which may impact intergroup differences and trend data.

- Introduction:

       - please stated the clear aim and add the references for the statement 'specialities with a greater percentage of female physicians
would have greater engagement with PCC and its derivative concepts of care.' (Line 93).

The authors have reviewed the line in question...

"As PCC focuses on the core values of empathy, respect, communication, and shared decision making [7,16,28], we hypothesize that specialties with a greater percentage of female physicians would have greater engagement with PCC and its derivative concepts of care." 

As the statement in question is a hypothesis, we do not believe it requires a citation. 

To clarify the study aim, the following sentence was added to the introduction:

This study aims to determine whether any correlations exist between the adoption of these concepts and changing gender ratios within specialties.

      - Did each speciality demonstrate the differences in PCC, PeCC, RCC and FCC? could it possible that the factor affecting the care be specialities instead of genders?

Differences in PCC, PeCC, RCC and FCC were statistically significant across the specialities and these differences were strongly correlated with the gender ratios of each specialty. 

      - would it be reliable to report google trends in a scientific paper?

The authors believe the use of Google Trends data, while not independently verifiable, is a novel way to look at the popularity of important topics that may have a significant impact on healthcare and its delivery.

The authors have added the following sentence to the limitation section:

Another limitation is the novel reporting of Google Trend data, the reliability of which is not independently verifiable.

Round 2

Reviewer 1 Report

The authors improved the manuscript and it can be accepted.